# *Ancyronyx jhoanae* sp. nov. (Coleoptera: Elmidae), A New Spider Riffle Beetle Species from Luzon, Philippines, and New Records for *A. tamaraw* Freitag, 2013 †

**Christalle Beatriz N. Seno** *⁠ and Hendrik Freitag

Ateneo Biodiversity Research Laboratory, Department of Biology, Ateneo de Manila University, Katipunan Avenue, Loyola Heights, Quezon City 1108, Metro Manila, Philippines
* Correspondence: christalle.seno@obf.ateneo.edu
† urn:lsid:zoobank.org:pub:C808256F-DB64-4C09-BCBB-C9EA6ACEA6DD,
  urn:lsid:zoobank.org:act:76007C04-F9FC-4554-A42E-441CE1E5F096.

**Abstract:** *Ancyronyx jhoanae* sp. nov., a new species of genus *Ancyronyx* Erichson, 1847 from Luzon is described using an integrative taxonomic approach. Illustrations of habitus and diagnostic characters are provided. Molecular analysis of a fragment of the COI 5'-end was employed to support the morphological species concept. Differences from closely related species based on molecular and morphological data are discussed. First records of *A. tamaraw* Freitag, 2013 from Luzon are reported.

**Keywords:** biogeography; biodiversity; COI; DNA barcoding; integrative taxonomy

## 1. Introduction

During the past decade, taxonomic surveys in major islands of the Philippine archipelago have raised the number of known spider riffle beetle species in the country from 10 in 2012 to 17 by 2022 [1–3]. The smaller-bodied *Ancyronyx patrolus* Freitag and Jäch, 2007 species group appears to be more speciose in the archipelago compared to the larger-bodied *Ancyronyx variegatus* (Germar, 1824) species group. Members of the former tend to be regionally endemic. In fact, more than half of the known species in the Philippines are currently listed with single-island distributions [1].

The recent descriptions of new species from southern Philippines by Seno et al. [1] point to a distribution potentially spanning the entire archipelago and emphasize the presence of undiscovered diversity in the taxon. Major islands tend to harbor more than one species, which often occur syntopically [2,4]. This underscores the necessity for further taxonomic studies and the application of integrative approaches to expedite species discovery especially considering the continued decline in insect biodiversity.

An analysis of morphological and molecular data from existing collections led to the description of a new species, *Ancyronyx jhoanae* sp. nov., and new records for *A. tamaraw* Freitag, 2013 from Luzon Island, Philippines. This study constitutes the first published record of the *Ancyronyx patrolus* species group on the island.

## 2. Materials and Methods

### 2.1. Taxon Sampling

Samples were collected from the central lowlands of Luzon island, located in the northern region of the Philippines. The island is part of the Greater Luzon intra-biogeographic region, one among seventeen subregions suggested to harbor distinct sets of flora and fauna due to the geologic history of the archipelago [5]. The specimens were manually captured, either by handpicking from submerged substrates or by hand nets. Samples were subsequently stored in 95% ethanol. Based on the label, some older material was collected from submerged wood and stored dry for almost 20 years. Specimens used in this study

were partly glued onto entomological cards and deposited in the following repositories: ADMU (Ateneo de Manila University, Quezon City, Philippines); NMW (Naturhistorisches Museum Wien, Wien, Austria); PNM (National Museum of the Philippines, Museum of Natural History, Manila, Philippines).

## 2.2. Morphological Studies

Examination of specimens was performed using an Olympus SZ61 dissecting microscope (Olympus, Tokyo, Japan) [1]. Selected male and female individuals were then dissected. Separated terminal abdominal parts and genitalia were mounted on microscope slides with 88.0% lactic acid [6]. An Olympus CX21 compound microscope and DinoEye Eyepiece camera (AnMo Electronics Corp., New Taipei City, Taiwan) were used to observe and photograph the prepared slides [1]. Adobe Illustrator 25 (Adobe, San Jose, USA) was utilized to produce vector illustrations of the male and female genitalia.

Selected specimens were measured using an eyepiece graticule. The following abbreviations are used in the description: CL (combined length of elytra and pronotum); EL (elytral length); EW (maximum elytral width); HW (head width); ID (interocular distance); MW (maximum pronotal width); PL (pronotal length).

A Canon EOS 6D camera with MP-E 65 mm f/2.8 Macro Photo lens (Canon, Tokyo, Japan) was used to obtain habitus photographs. Subsequent stacking of image series was performed through Helicon Focus 7.6.1 (Helicon Soft, Kharkiv, Ukraine) [1]. Textual description of external morphology and genitalia followed terminology from the Handbook of Zoology by Kodada et al. [7].

## 2.3. Molecular Species Delimitation

After dissection, DNA was obtained from selected individuals using Qiagen DNeasy kit (Qiagen, Hilden, Germany). Extraction protocols in Qiagen [8] were followed. A fragment of the cytochrome c oxidase I (COI) gene 5′-end was amplified using primer pair LCO1490 (5′-GGT CAA CAA ATC ATA AAG ATA TTG G-3′) and HCO2198 (5′-TAA ACT TCA GGG TGA CCA AAA AAT CA-3′) [9]. The PCR procedure utilized the 2× Taq Master Mix (Vivantis), following the thermocycler profile described by Freitag [10]. Cleaning and sequencing of amplicons were performed as a service by Macrogen Europe Inc. (Amsterdam, The Netherlands).

Chromatogram analysis and manual trimming of ambiguous ends was conducted using MEGA11 [11]. COI sequences of other *Ancyronyx* species from other Philippine islands and Borneo were obtained from Genbank (Table 1). The COI sequence of *Grouvellinus quest* Freitag, Pangantihon & Njunjić, 2018 from Genbank was used as an outgroup. Sequences were aligned using built-in MUSCLE [12] in MEGA11 [11].

**Table 1.** GenBank accession numbers and specimen geographical origins of *Ancyronyx* and *Grouvellinus* sequences used in the COI sequence analysis.

| Species | Locality | Voucher | GenBank No. |
|---|---|---|---|
| *A. berghaueri* Sabordo, Delocado & Freitag, 2020 [2] | Negros | FR569 | MT568726 |
| *A. clisteri* Kodada et al., 2020 [13] | Borneo | FZ1640 | MK505421 |
| *A. kalasan* Seno, Delocado & Freitag, 2022 [1] | Mindanao | FR337 | OP609914 |
| *A. manobo* Seno, Delocado & Freitag, 2022 [1] | Mindanao | BS017 | OP609918 |
| *A. minerva* Freitag & Jäch, 2007 [4] | Palawan | FR467 | MT568723 |
| *A. punkti* Freitag & Jäch, 2007 [4] | Palawan | FR466 | MT568722 |
| *A. schillhammeri* Jäch, 1994 [14] | Mindanao | FR380 | OP609913 |
| *A. jhoanae* Seno & Freitag, sp. nov. | Luzon | FR238 | OQ259604 |
| | | FR338 | OQ259605 |
| *A. tamaraw* Freitag, 2013 [10] | Luzon | FR271 | MT568727 |
| *A. zamboangaensis* Seno, Delocado & Freitag, 2022 [1] | Mindanao | FR387 | OP609911 |
| *G. quest* Freitag, Pangantihon & Njunjić, 2018 [15] | Borneo | H17 | LR738834 |

A maximum likelihood (ML) tree with 1000 bootstrap replicates was generated using MEGA11 [11]. General time reversible (GTR + G + I) was identified through AICc as the best substitution model. Molecular species delimitation was implemented by employing three methods. First, the generated tree was used as an input for Poisson Tree Processes (PTP) [16] via an online portal (https://mptp.h-its.org/#/tree; accessed on 25 November 2022). Second, the COI sequence alignment was used to determine putative species clusters through a webserver (https://bioinfo.mnhn.fr/abi/public/asap/; accessed on 25 November 2022) that runs Assemble Species by Automatic Partitioning (ASAP) [17]. Third, pairwise intra- and interspecific genetic divergences of sequences were computed through uncorrected *p*-distance in MEGA11 [11]. Putative species clusters were determined from the computation using a 3.00% threshold.

## 3. Results

### 3.1. DNA Analyses

The alignment of COI 5′ sequences generated a matrix 662 bp long with no insertion-deletions, gaps, or ambiguous sites. Maximum likelihood (ML) analysis retrieved *Ancyronyx jhoanae* sp. nov. as a strongly supported clade within a complex encompassing species from Negros (*A. berghaueri*) and Mindanao (*A. kalasan*, *A. manobo*, *A. zamboangaensis*) islands (Figure 1). The interspecific distance of *A. jhoanae* sp. nov. with other *Ancyronyx* species in the analysis ranges from 4.38% to 14.80%, with the lowest genetic distance from *A. berghaueri*. Comparatively, intraspecific distance is low at 1.21% (Table 2).

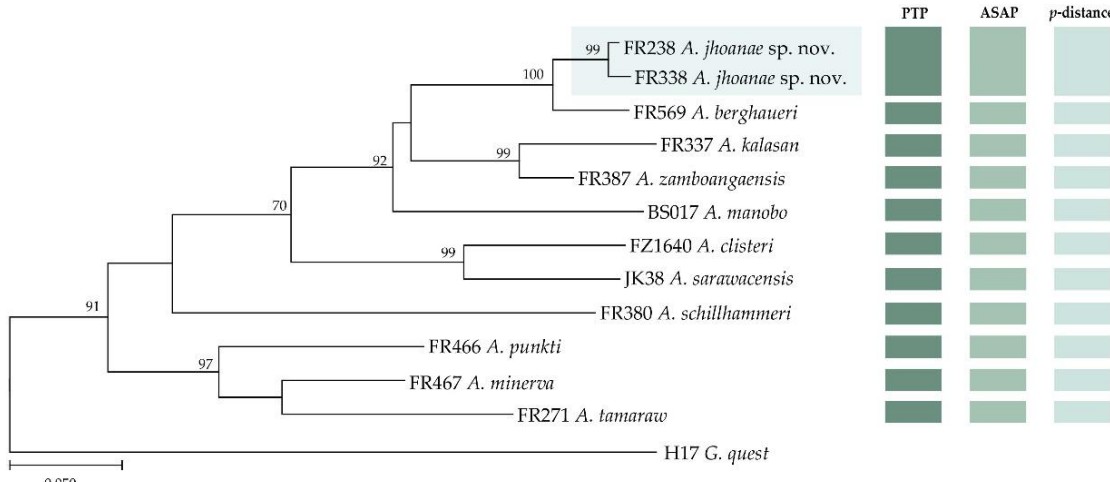

**Figure 1.** Molecular species delimitation in selected *Ancyronyx* species based on COI alignment (662 bp) with putative species clusters determined through Poisson Tree Processes (PTP), Assemble Species by Automatic Partitioning (ASAP), and *p*-distance (*p*). Maximum likelihood (ML) tree shows topology based on general time-reversible (GTR + G + I) model, 1000 bootstraps. Bootstrap values below 70 are not shown.

**Table 2.** Uncorrected *p*-distance (in %) of *Ancyronyx* species based on COI alignment (662 bp).

| # | Species | 1 | 2 | 3 | 4 | 5 | 6 | 7 | 8 | 9 | 10 | 11 | 12 |
|---|---------|---|---|---|---|---|---|---|---|---|----|----|----|
| 1 | *A. jhoanae* sp. nov. | | | | | | | | | | | | |
| 2 | *A. jhoanae* sp. nov. | 1.21 | | | | | | | | | | | |
| 3 | *A. berghaueri* | 4.38 | 4.98 | | | | | | | | | | |
| 4 | *A. clisteri* | 11.18 | 11.03 | 11.33 | | | | | | | | | |
| 5 | *A. kalasan* | 9.21 | 9.82 | 9.06 | 12.54 | | | | | | | | |
| 6 | *A. manobo* | 10.57 | 10.12 | 10.12 | 12.84 | 11.48 | | | | | | | |
| 7 | *A. minerva* | 12.99 | 12.84 | 12.08 | 13.29 | 13.90 | 13.44 | | | | | | |
| 8 | *A. punkti* | 13.60 | 13.14 | 13.29 | 14.35 | 15.41 | 13.90 | 8.46 | | | | | |
| 9 | *A. sarawacensis* | 11.48 | 11.33 | 11.78 | 8.76 | 12.99 | 13.29 | 13.90 | 13.29 | 13.90 | | | |
| 10 | *A. schillhammeri* | 14.80 | 14.65 | 15.11 | 13.60 | 14.65 | 15.11 | 14.05 | 10.42 | 13.44 | 15.11 | | |
| 11 | *A. tamaraw* | 12.39 | 12.54 | 12.69 | 13.44 | 12.08 | 14.05 | 8.46 | 14.35 | 11.48 | 14.20 | 13.29 | |
| 12 | *A. zamboangaensis* | 9.67 | 9.06 | 9.37 | 12.39 | 6.19 | 9.67 | 13.60 | 15.56 | 17.98 | 18.13 | 17.52 | 16.62 |

*3.2. Taxonomy*

*Ancyronyx patrolus* species group (sensu Freitag & Jäch 2007)
*Ancyronyx jhoanae* sp. nov.
urn:lsid:zoobank.org:act:76007C04-F9FC-4554-A42E-441CE1E5F096

**Holotype**. ♂[FR238]: Philippines, (PNM) "PHIL: Luzon, Nueva Vizcaya, Quezon, / Villaverde, Kisikis River, sec. veget./rural; / 266 m a.s.l.; 16°34′21″ N 121°9′58″ E; / 31 Jul 2018, leg. Garces & Cagande (439)"; terminal abdominal parts dissected and glued separately.

**Paratypes.** Philippines, Luzon: 3♂♂, 3♀♀: (ADMU) same data as holotype; 9♂♂[FR338], 7♀♀: (ADMU) Nueva Vizcaya, Quezon, Solano, San Juan River trib., Mangilocos Creek, sec. veget./rural; 270 m a.s.l.; 16°34′15″ N 121°09′43″ E; 31 July 2018, leg. Garces & Cagande (438); 3♂♂, 1♀; (NMW, ADMU) Aurora, San Luis, Brgy. Ditumabo, Mother falls, run, submerged wood; 160 m a.s.l.; 15°41′10″ N 121°28′27″ E; 11 February 2003, leg. Mendoza (M466).

**Adult.** In addition to the male holotype, one male and two female paratypes were measured. Body: CL: 1.26–1.33 mm (1.30 mm); EW: 0.62–0.69 mm (0.65 mm); CL/EW: 1.93–2.11 (1.99).

**Coloration** (Figure 2A): head capsule, mouthparts, pronotum, scutellum and elytra (except for two pairs of yellow patches) dark brown to black; anterior yellow elytral patches oblique, approx. as wide as long, subtrapezoidal, extending from humeri approx. up to second row of elytral punctures, not reaching elytral suture and scutellum; posterior yellow elytral patches longitudinal, subtriangular, extending up to second row of punctures not reaching lateral margins, suture, nor apex; legs predominantly yellow; coxae, trochanter, and tarsomeres 1–4 brownish; femorotibial articulations and distal end of tarsomere 5 dark brown; claws brown at base, increasingly yellowish distally; antennae yellow except for scapus and terminal segment; ventral side predominantly yellow, darker in older specimens.

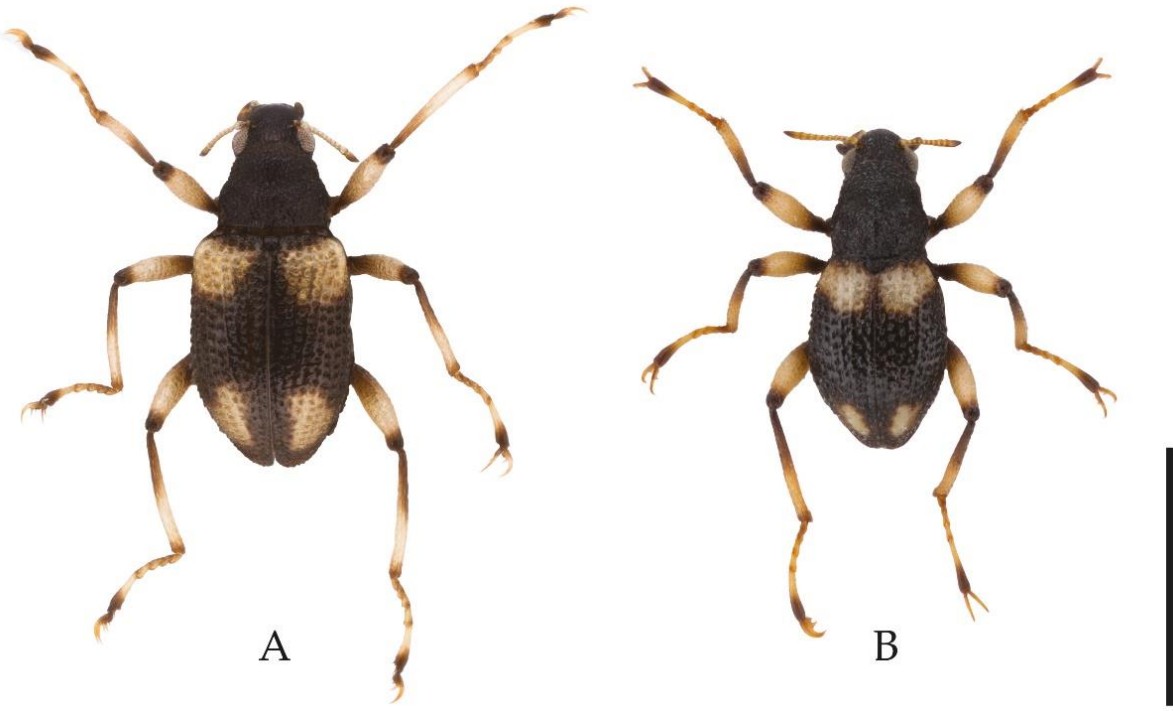

**Figure 2.** Dorsal habitus images of (**A**) *Ancyronyx jhoanae* sp. nov. and (**B**) *A. tamaraw* Freitag, 2013.

**Head** (Figure 3A,B): HW: 0.26–0.29 mm (0.27 mm); ID: 0.19–0.20 mm (0.19 mm); labrum smooth with sparse, trichoid setae; clypeus and frons reticulate with longitudinal striae; frontoclypeal suture convex and conspicuous. Eyes moderately protruding. Anten-

nae (Figure 3A) with 11 antennomeres, slender, slightly longer than head width. Genae reticulate with sparse pubescence. Gula microstriate; gular sutures distinct.

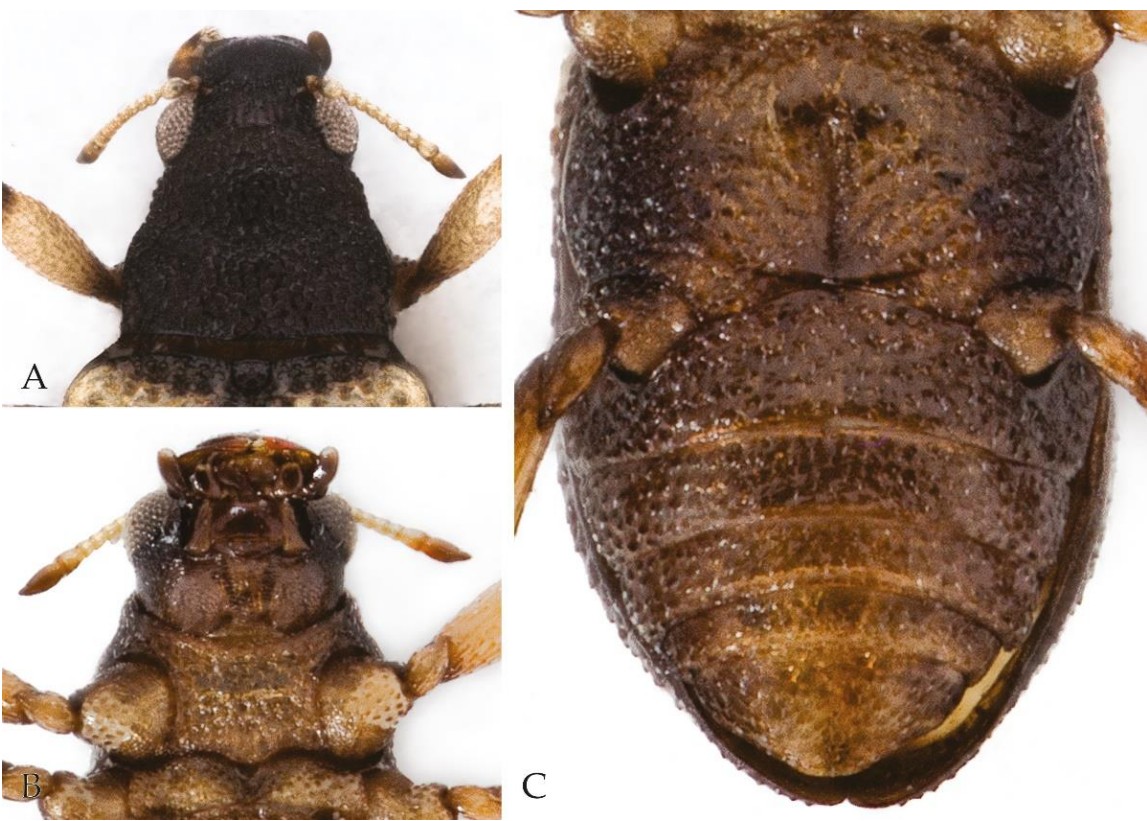

**Figure 3.** *Ancyronyx jhoanae* sp. nov.: (**A**) anterodorsal aspect with head and pronotum; (**B**) anteroventral aspect with head, prosternum, and mesoventrite; (**C**) and posteroventral aspect with metaventrite and ventrites 1–5.

**Thorax.** Pronotum (Figure 3A) PL: 0.34–0.36 mm (0.35 mm); MW: 0.42–0.47 mm (0.44 mm); wider than long (PL/MW: 0.7–0.9), widest at about posterior 0.3, narrower than elytra; transverse groove deep; area posterior to transverse groove distinctly more elevated than anterior area; posterolateral oblique grooves shallower than transverse groove but distinct; lateral margin bisinuous; anterior margin weakly convex; posterior margin almost straight, with slight grooves in area adjacent to scutellum edges; pronotal surface bipunctate with dense micropunctures superimposing large, shallow punctures; lateral pronotal carina conspicuous only in posterior 0.3. Prosternum moderately densely punctate with short, sparse pubescence; prosternal process subpentagonal, wider than long, slightly impressed posteromedially; posterior margin slightly bisinuous. Metascutellum (Figure 3A) subtriangular, slightly rugose. Elytra (Figure 2A) elongate; EL 0.89–1.00 mm (0.94 mm); EW 0.62–0.67 mm (0.65 mm); ca. 1.4 times as long as wide (EL/EW), widest at about midlength; with ten longitudinal, moderately impressed rows of punctures; six rows between suture and humerus; punctures somewhat regularly arranged, large and moderately impressed on disc; punctures on elytral declivity denser, smaller, and shallower; interstices and intervals slightly rugose near suture especially in basal half, glabrous towards lateral margins; short trichoid setae sparsely scattered across elytra, rarely with a few long setae; lateral elytral gutter narrow; humeri rounded and moderately prominent; elytral apices separately rounded. Mesoventrite short, moderately densely punctate, with short, sparse pubescence; median cavity moderately impressed; discrimen inconspicuous; paired posterolateral impressions shallow. Metaventrite slightly longer than combined length of pro- and mesoventrite; metaventral discrimen deeply impressed, glabrous only in

most posterior areas. Hind wings fully developed. Legs (Figure 2A) slightly shorter than the body; coxae large, obliquely conoidal; only procoxae visible in dorsal view; trochanter short, lanceolate; femora and tibiae longitudinally striated by elongate tubercles; distal end of tibiae with few, short setae; distal end of tarsomere 5 with one or two long, conspicuous setae; claws large, slender, bent, each with three teeth; most distal tooth significantly larger than inconspicuous basal tooth.

**Abdomen.** Ventrite 1 slightly projected anteriorly, rugose, with small, setiferous tubercles; surface of ventrites 2–4 similar to ventrite 1, setiferous tubercles smaller distally; ventrite 2 longer than 3 and 4; ventrite 3 and 4 combined length slightly shorter than ventrite 1; ventrite 5 slightly shorter than ventrite 1; in male, broadly semicircular (ca. 160 µm long, 320 µm wide); in female, broadly subtriangular (ca. 240 µm long, 360 µm wide). Male sternite IX (Figure 4D) ca. 315 µm long, apical corners broad, rounded; lateroapical surface with sparse, short setae; apical margin weakly concave, with very few long setae; paraprocts not reaching apical margin. Female sternite VIII (Figure 4E) ca. 330 µm long; ventromedian surface densely pubescent; median strut long, slightly widened apicad; lateroposterior margins with moderately long setae. Tergite VIII in male semicircular, wider than long (ca. 150 µm long, 190 µm wide), increasingly reticulate apicad; condyles rounded, conspicuous; in female subpentagonal, wider than long (ca. 170 µm long, 220 µm wide), reticulate; condyles small, conspicuous.

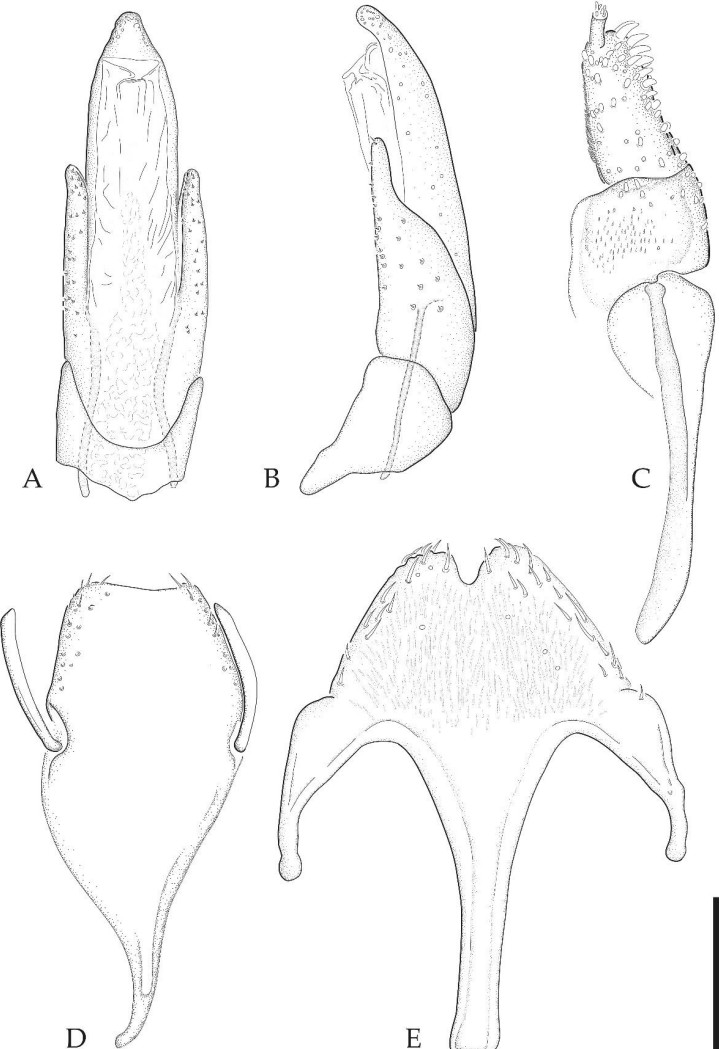

**Figure 4.** Vector illustrations of *Ancyronyx jhoanae* sp. nov.: (**A**) aedeagus, ventral view; (**B**); aedeagus, lateral view; (**C**) ovipositor; (**D**) male sternite IX; and (**E**) female sternite VIII. Scale bar = 100 µm.

**Aedeagus** (Figure 4A,B) ca. 310 μm long; median lobe long, moderately slender, conical towards broadly subtriangular, rounded apex (Figure 4A), slightly curved ventrad (lateral view, Figure 4B), with pores most dense at apex; ventral sac plicate, weakly sclerotized, not reaching apex (Figure 4A,B); fibula weakly sclerotized, inconspicuous; corona inconspicuous. Phallobase asymmetrical, longer ventrally (lateral view, Figure 4B), with weakly sclerotized basal margins; basolateral (penile) apophyses slender, slightly protruding out of phallobase; ejaculatory duct inconspicuous. Parameres short, reaching about basal 0.7 of aedeagus, subtriangular strongly tapering from apical 0.3 to 0.6 then extending subparallel into a narrow, rounded apex, with sparse, short setae most densely arranged near ventral margin; basal margin oblique, not emarginate (lateral view, Figure 4B).

**Ovipositor** (Figure 4C) ca. 410 μm long. Stylus short, slightly bent outwards, with about three subequal moderately long and some short inconspicuous apical sensilla. Coxite moderately stout, with large, peg-like spines in lateroapical surface; spines sparser and short cuboid medially and basad; inner margin extendedly densely pubescent with short, trichoid setae. Valvifer longer than coxite; longitudinal baculum slightly curved inwards; narrowest at median portion.

**Differential Diagnosis.** *Ancyronyx jhoanae* sp. nov. most closely resembles *A. berghaueri* from Negros island but differs in the following characteristics: (a) a shorter aedeagus ca. 310 μm vs. 360 μm in *A. berghaueri* (Figure 4A,B vs. Sabordo et al. 2020: Figs. 9–10 [2]); (b) a distinct paramere shape that is primarily subtriangular but sharply tapers into a narrow extension with subparallel margins (Figure 4B) vs. subtriangular with slightly bent, oblique conical apical extension in *A. berghaueri* (see Sabordo et al. 2020: Figs. 9–10 [2]); and (c) a longer ovipositor of ca. 410 μm length with short cuboid spines medially (Figure 4C) vs. a 370 μm long ovipositor with pointed peg-like spines medially in *A. berghaueri* (see Sabordo et al. 2020: Fig. 11 [2]).

**Etymology.** The new species is named after Dr. Jhoana Garces of the Ateneo Biodiversity Research Laboratory to recognize her efforts as the collector of some of the specimens examined in this study and to honor her contributions to freshwater macroinvertebrate research in the Philippines.

**Distribution.** *Ancyronyx jhoanae* sp. nov. is currently only known to occur in Nueva Vizcaya and the Aurora province located in the midwestern portion of Luzon island, Philippines.

*Ancyronyx tamaraw* Freitag, 2013

**Material Examined.** Philippines, Luzon: 1♂, 2♀♀[FR271]: (ADMU) Subic Bay Freeport Zone (SBFZ), Binictican River, upper forested, bottom gravel; 30 m a.s.l.; 14°48′22″ N 120°19′50″ E; 01 Jul 2017, leg. Freitag (SBiR3c); 1♂, 4♀♀: (ADMU) Subic Bay Freeport Zone (SBFZ), upper El Kabayo trib., small hill str.; 80 m a.s.l.; 14°47′58″ N 120°19′49″ E; 25 June 2017, leg. Freitag (SER3); 2♂♂: (ADMU) Subic Bay Freeport Zone (SBFZ), Triboa River, small hill str., sec. forest, upstr. dam, riffle, rock surface; 65 m a.s.l. 14°46′16″ N 120°17′58″ E; 14 April 2018, leg. Freitag (Sub1g).

**Remarks.** Individuals from the Subic Bay area (Bataan Peninsula) generally tend to have slightly larger anterior elytral patches than specimens from the type locality in Mindoro. In some specimens, the yellow anterior patches extend to the elytral suture.

**Distribution.** *Ancyronyx tamaraw* was previously only recorded from the island of Mindoro (Tamaraw Falls, Puerto Galera). Here it is reported to occur in the Subic Bay Freeport Zone (SBFZ) located in the Bataan Peninsula at the western edge of Luzon island, Philippines.

## 4. Discussion

This study provides a description of the first *Ancyronyx* species endemic to the island of Luzon (Figure 5). Given previous observations in Sulawesi [6] and Mindanao [1], several species of this lineage appear to resemble each other externally, as observed here for *A. jhoanae* sp. nov. and *A. berghaueri*. Although the genus has been known for vivid coloration and species-specific markings, it appears to be not uncommon that similarity in

color patterns render some species almost indistinguishable from each other in terms of external morphology.

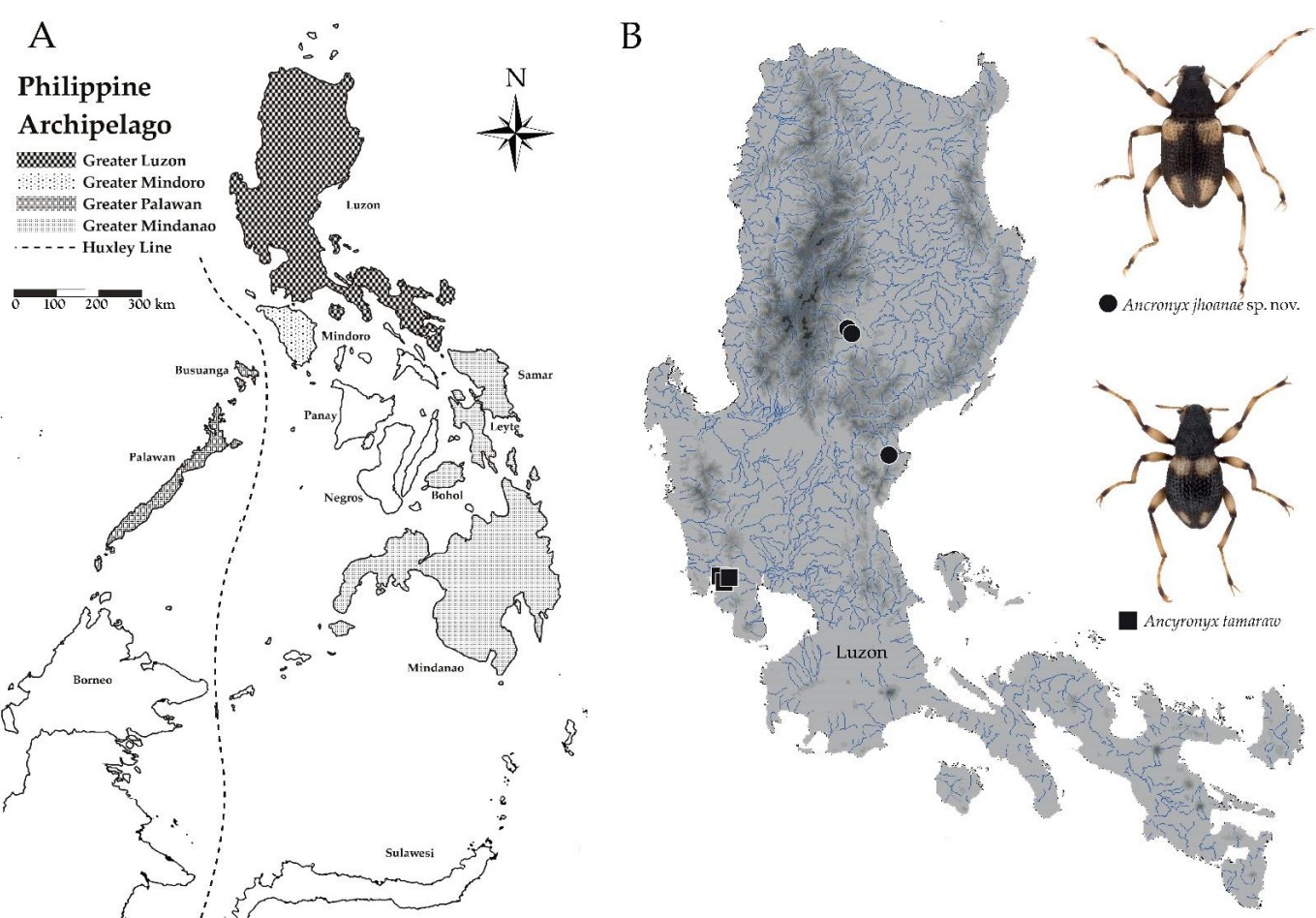

**Figure 5.** Maps with (**A**) major intra-Philippine biogeographical regions and (**B**) collection sites of *Ancyronyx jhoanae* sp. nov. and *A. tamaraw* Freitag, 2013 in Luzon.

Molecular data suggest that the newly identified species is part of a lineage previously identified by Seno et al. [1]. This lineage encompasses species from Greater Mindoro, Greater Mindanao, and Sulawesi. Together with *A. jhoanae* sp. nov., these species constitute a subgroup genetically and geographically distinct from those of Greater Palawan where the namesake species and subgroup of the *A. patrolus* species group is known to occur [4]. Both clades appear to be separated by the Huxley Line (Figure 5A), although a Sundaic origin for the *A. patrolus* subgroup is currently not evident due to the lack of related representatives in Borneo and other Sundaic islands.

In contrast, representatives of the *A. minerva* subgroup (sensu Freitag & Jäch [4]) occur on both sides of the Huxley Line in the Philippines (Figure 5A). *Ancyronyx minerva* Freitag & Jäch, 2007 is known from Busuanga, Mindoro, and Palawan [4,8]. Now, the distribution area of the related *A. tamaraw* has also been extended and covers more than just one intra-Philippine biographic region [5]: Greater Mindoro and Greater Luzon (Figure 5A). Presumed records from Bohol [18] still require confirmation by more detailed study.

**Author Contributions:** Conceptualization, methodology, investigation, data curation, formal analysis, writing—review and editing, C.B.N.S. and H.F.; software, writing—original draft preparation, visualization, C.B.N.S.; validation, resources, supervision, project administration, funding acquisition, H.F. All authors have read and agreed to the published version of the manuscript.

**Funding:** This research was funded by University Research Council Grant (URC 2022-09) of Ateneo de Manila University, Philippines.

**Institutional Review Board Statement:** Not applicable.

**Informed Consent Statement:** Not applicable.

**Data Availability Statement:** Not applicable.

**Acknowledgments:** We would like to thank Clister Pangantihon for his assistance in the laboratory. Our gratitude goes out to the Local Government Unit (LGU) of Quezon, Nueva Vizcaya, and Subic Bay Metropolitan Authority (SBMA), Zambales, whose kind assistance and cooperation have allowed laboratory members to conduct sampling of specimens under the Gratuitous Permit (GP 0133-17) issued by the Bureau of Fisheries and Aquatic Resources (BFAR). We also express our gratitude to the Biodiversity Teaching in a Philippine-Cambodian-German Network (BIO-PHIL) project funded by the German Academic Exchange Service (DAAD project BIO-PHIL 57393541) for their support to the research activities of the Ateneo Biodiversity Research Laboratory.

**Conflicts of Interest:** The authors declare no conflict of interest. The funders had no role in the design of the study; in the collection, analyses, or interpretation of data; in the writing of the manuscript, or in the decision to publish the results.

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
