# Peer review of "Ancyronyx jhoanae sp. nov. (Coleoptera: Elmidae), A New Spider Riffle Beetle Species from Luzon, Philippines, and New Records for A. tamaraw Freitag, 2013"

_2673-6500, doi:10.3390/taxonomy3010008_

Round 1

Reviewer 1 Report

I agree that the species described in this paper is a new species . The morphology is well described, and the DNA barcode information is very useful.

The photo in Fig. 3 is too dark to clearly see the punctures mentioned in the description. It needs to be improved.

A key to species is required for a region (e.g.Philippines ) or species group.

Author Response

Thank you for the feedback.

  • The photo in Fig. 3 has been edited. 
  • A key was not included in this publication because (1) it was thought to be more suitable to include a Philippine key in an upcoming publication revising species groups, and (2) several Ancyronyx species from northern Luzon are probably still awaiting description (more material needed), rendering any key included in this publication would likely be more temporary than practical.
  • Figure 5 was edited 

Reviewer 2 Report

The manuscript is well written and only minor corrections are required. All my comments and suggestions are included in the attached PDF file.

Author Response

Thank you for the feedback.

Revisions were made according to the reviewer's suggestions.

Reviewer 3 Report

In the material and method, sufficient information about the study area has not been shared. The Philippines should be mentioned in detail. Because in the discussion part, it was emphasized to which subgroup the new species is associated and its distribution within the country.

There are some typographical errors. Needs to be fixed.

It would be more accurate to include actual pictures of aedeagus as they are an important character in the identification of the species.

Author Response

Thank you for the feedback.

  • Revisions were made in the material and methods and discussion to provide more information regarding the study area.
  • Typographical error and comments are addressed through comment replies in the attached file.
  • Aedeagal images were not included in this publication because the three-dimensional quality of the aedeagus is usually better captured by scientific illustrations. Illustrations are improved in the revised version as there was a rendering problem before.
  • Figure 5 was adjusted, but we refrain from indicating provinces, since biogeographic units like islands, river systems, mountains matter, but not (artificial) political province borders. Nevertheless, the respective provinces can be retrieved from the specimen labels
